# BlockFound: Customized blockchain foundation model for anomaly detection

## Abstract

We propose `BlockFound`, a customized foundation model for anomaly blockchain transaction detection. Unlike existing methods that rely on rule-based systems or directly apply off-the-shelf large language models, `BlockFound` introduces a series of customized designs to model the unique data structure of blockchain transactions. First, a blockchain transaction is multi-modal, containing blockchain-specific tokens, texts, and numbers. We design a modularized tokenizer to handle these multi-modal inputs, balancing the information across different modalities. Second, we design a customized mask language learning mechanism for pretraining with RoPE embedding and FlashAttention for handling longer sequences. After training the foundation model, we further design a novel detection method for anomaly detection. Extensive evaluations on Ethereum and Solana transactions demonstrate `BlockFound`'s exceptional capability in anomaly detection while maintaining a low false positive rate. Remarkably, `BlockFound` is the only method that successfully detects anomalous transactions on Solana with high accuracy, whereas all other approaches achieved very low or zero detection recall scores. This work not only provides new foundation models for blockchain but also sets a new benchmark for applying LLMs in blockchain data.

## 1 Introduction

With the rapid development of blockchain technology, cryptocurrencies have gained significant attention and are increasingly being used in real-world applications. A lot of Decentralized Finance (DeFi) protocols have emerged, offering a wide range of financial services, such as lending, borrowing, and trading, to users. However, the decentralized nature of these protocols also makes them vulnerable to various security threats, including the presence of malicious attacks such as double-spending attack (Karame et al., 2012), partition attacks (Saad et al., 2019), and front-running attacks (Eskandari et al., 2020). These attacks seriously threaten the asset security of billions of blockchain users. For example, at least 3.24 billion USD were lost to DeFi attacks from April 2018 to April 2022 (Werner et al., 2022).

Having a runtime anomalous transaction detection is critical for protecting user assets. Such systems aim to detect suspicious transactions that deviate from typical patterns and provide early warnings of potential security threats, enabling quick interventions to minimize the impact of malicious activities (Ravazzi, 2024). Moreover, as the complexity and volume of transactions in DeFi continue to grow, robust anomaly detection mechanisms will become increasingly essential to ensure the long-term stability and security of blockchain-based financial services.

Prior works in detecting anomalous transactions primarily rely on heuristic-based approaches, which examine features like transaction time, input addresses, and output addresses. However, these heuristic methods have limited generalizability in that they cannot detect attacks that do not follow pre-defined patterns. Given that attacks often change rapidly and summarizing rules requires extensive expertise and effort, these rule-based methods cannot fulfill the requirements of detecting modern blockchain attacks. To enable better generalizability and less human effort, some recent works seek data-driven methods that leverage machine learning models such as Gaussian Mixture Models (GMM) (Yang et al., 2019) and Long Short-Term Memory (LSTM) networks (Aldaham & HAMDI, 2024) to learn normal transaction patterns and conduct anomaly detection based on the learning

models. However, such traditional and small models have limited capacity to digest large-scale datasets and thus are again limited in generalizability. Besides, these models cannot capture the long-range dependencies and complex temporal dynamics inherent in transaction data (Parisotto et al., 2020; Zeyer et al., 2019; Wen et al., 2022), resulting in sub-optimal modeling performance. Motivated by the success of Transformer-based foundation models (Devlin et al., 2019; Liu et al., 2019; Achiam et al., 2023) in many other domains. recent research (Gai et al., 2023) also uses the off-the-shelf transformer-based model, GPT to train blockchain transactions and conduct anomaly detection. As we will show in §5, without modeling the unique data structure of blockchain transactions, this GPT-based approach (BlockGPT) achieves only limited anomaly detection performance.

In this work, we propose `BlockFound`, a customized foundation model for detecting anomalous transactions in DeFi. Technically speaking, we still follow BlockGPT and use a transformer as our foundation model. However, rather than using GPT-style models, we select BERT and use mask language modeling (MLM) to train the foundation model due to their suitability for the task. Our primary objective is to learn meaningful representations of transaction patterns to identify anomalies effectively, rather than to generate new sequences. While GPT-style models are powerful for generative tasks, MLMs offer an effective framework for encoding input data without the necessity of learning autoregressive sequence generation. This makes MLMs particularly well-suited for our anomaly detection task. Given that a transaction is multi-modal, containing blockchain-specific tokens, texts, and numbers, we design a novel tokenizer that integrates different tokenization strategies for different types of inputs. Further, we leverage RoPE and FlashAttention to modify the base BERT model such that our foundation model can handle long inputs. We train our foundation model on a large dataset of benign transactions to learn the patterns of normal transactions, then feed the trained model with masked transactions and calculate how well the model can reconstruct the transaction. We use the reconstruction errors as the metric for anomaly detection. Given the model learns and reconstructs transactions based on normal transaction patterns, a transaction that is difficult to predict, is more likely to be abnormal.

We evaluate the performance of `BlockFound` on real-world transactions collected from the Ethereum and Solana networks. We compare `BlockFound` with four baselines: a rule-based approach, a traditional ML-based approach, BlockGPT, and a method that we directly query GPT-4o if an input transaction is anomalous. We show that `BlockFound` can detect anomalous transactions with high accuracy and a low false positive rate, significantly outperforming existing methods. We further conduct detailed ablation studies to verify our key design choices, including the tokenizer and our methods for handling long inputs. Finally, we evaluate the sensitivity of our method to key hyper-parameters, such as the token mask percentage and model structure. We open-source the code, model, and datasets used in the anonymous link [1]. To the best of our knowledge, `BlockFound` is the first work to provide a low FP detection method and an open-source implementation for LLM-based anomalous transaction detection in DeFi. We hope that our approach can serve as a useful tool for protecting user assets in DeFi, and that our dataset and implementation can train future researchers in this critical area.

## 2 BACKGROUND

**Blockchain** Blockchain is a decentralized, distributed ledger technology designed to enable secure, transparent, and tamper-resistant record-keeping. Originally developed as the backbone of cryptocurrencies like Bitcoin Nakamoto (2008), blockchain consists of a chain of blocks, each containing a list of transactions. The core innovation of blockchain is its ability to achieve consensus across a network of nodes without relying on a central authority. This is accomplished through consensus mechanisms such as Proof of Work (PoW) Back et al. (2002) or Proof of Stake (PoS) Saleh (2021), which ensure that all participating nodes agree on the state of the ledger. Blockchain's immutability and transparency are crucial features that make it suitable for a variety of applications beyond cryptocurrencies, including supply chain management, healthcare, and finance. There are some well-known blockchain platforms, such as Bitcoin, Ethereum, and Solana.

**Smart Contracts and Transactions** Smart contracts are self-executing agreements where the terms are directly written into code, enabling automated and secure transactions. These contracts

---

[1]https://shorturl.at/9dFL1

are a fundamental component of decentralized applications (DApps), which run on peer-to-peer networks using blockchain technology to create systems that are secure, transparent, and resistant to censorship. Deployed on platforms like Ethereum, DApps facilitate more complex, programmable transactions beyond simple value transfers. Each blockchain transaction can trigger the execution of a smart contract, which autonomously processes conditions, manages assets, and updates the ledger. A typical transaction includes details such as the sender and recipient addresses, the amount of cryptocurrency or tokens transferred, and any data required to execute smart contracts. This automation enhances transparency, security, and trust within decentralized applications, making blockchain an ideal infrastructure for DeFi, gaming platforms, and supply chain management systems. However, smart contracts are immutable once deployed, meaning that any errors or vulnerabilities in the contract code can lead to significant risks, including financial loss. As a result, detecting anomalous transactions within smart contracts is crucial for maintaining the security and reliability of blockchain-based systems. In this work, we focus on detecting anomalous transactions, which deviate from typical patterns observed in benign transactions that can be associated with smart contract vulnerabilities at both the logic and implementation levels. Examples include transactions with abnormal method calls or sequences that differ significantly from expected patterns. Detecting such anomalous transactions within smart contracts is crucial for maintaining the security and reliability of blockchain-based systems, as it enables stopping potentially risky contracts to prevent losses when malicious behavior is detected (Hasan et al., 2024; Hassan et al., 2022).

## 3 EXISTING TECHNIQUES AND LIMITATIONS

**LLM-based detection.** Recently, a study (Gai et al., 2023) has utilized a large language model to detect anomalous transactions. Specifically, it adopts a GPT-like causal language modeling approach, training the LLM by predicting the next token in the transaction trace. Anomalous transactions are detected by ranking scores based on the log-likelihood of the predicted trace. However, this approach faces several fundamental limitations. Firstly, unlike natural language, transactions do not naturally form sequential data, making the prediction of the next token less meaningful for transaction traces. Moreover, the tokenization method used in the study is suboptimal, *e.g.,* numerical values such as transaction fees are rounded to avoid vocabulary explosion, potentially obscuring critical transaction details. Effective tokenization is crucial for the successful application of LLMs in this context, as it directly impacts both the representation of smart contracts and the sequence length of processed transactions. In addition to developing specialized language models for blockchain data, some approaches (Chen et al., 2023a) directly adopt existing language models (*e.g.,* ChatGPT) without further fine-tuning. These methods involve feeding ChatGPT with raw input transactions (*e.g.,* corresponding JSON files) and are limited by the maximum input length of the model and knowledge of the model.

**Rule-based and traditional ML-based approaches.** In contrast to LLM-based approaches, non-LLM methods for anomalous transaction detection can be grouped into two main categories: traditional machine learning-based and heuristic-based approaches. The first category applies conventional machine learning models, such as Gaussian mixture models, to estimate the density of input transactions (Yang et al., 2019). Transactions with lower density scores are flagged as potentially anomalous. However, these methods are highly dependent on the quality and expressiveness of the transaction features used to generate hidden representations, limiting their generalizability. The second category consists of heuristic-based techniques. For instance, one method suggests detecting anomalous transactions by analyzing sequence length (Gai et al., 2023), under the assumption that shorter transactions are more likely to be benign. However, this assumption is overly simplistic and flawed, as will be demonstrated in §5.2. Heuristic-based methods often suffer from being too rigid and can be easily bypassed by adversaries who do not conform to such patterns.

## 4 KEY TECHNIQUES

In this section, we first provide the overview of key techniques and then introduce them in detail. The pseudo algorithm can be found in Appendix A.

### 4.1 TECHNIQUE OVERVIEW

**Tokenizer.** As demonstrated in Figure 1, a blockchain transaction mainly consists of three types of inputs: function and address signature in hash values, function logs in natural languages, and function arguments in numbers. This hybrid data type makes a transaction naturally to be multi-modal. As such, directly applying existing tokenizers designed for language models to blockchain transactions will be problematic. First, existing tokenizers will treat hash values as numbers and divide them into sub-tokens. However, these numbers themselves are meaningless, instead, they are just used to represent different entities. Second, the numbers in blockchain transactions have a very large value range and large values frequently show up. Directly applying the existing tokenizers will divide a large number into many sub-tokens and thus result in ultra-long sequences for individual transactions. To solve the first issue, we use one-hot tokenization for hash values. We only consider the top 7,000 frequent hash values in our training dataset and treat the rest as "OOV" (Out of Vocabulary). This method can constrain the vocabulary size, which helps reduce model parameters and improve training efficiency. We further train our own number tokenization model to handle numbers. Different from existing tokenizers, our model can better tailor to the large numbers in blockchain transactions and give shorter token sequences for large numbers. Finally, we still apply the text tokenizer to function logs to capture their semantic meanings. As demonstrated in §5, our customized tokenizer is critical for learning foundation models and final anomaly detection.

**Model design.** We make a different design choice from BlockGPT (Gai et al., 2023) and use a BERT structure together with MLM for our foundation model. The key rationale is to reduce training complexity and improve overall training efficiency. Specifically, we do not need to generate new transactions, and training GPT models are in general more difficult than BERT as predicting the future without any context is harder than filling missing parts with certain context. Besides, our main focus is to learn patterns of normal transactions. As such, we select BERT with MLM, which provides enough pattern-learning capabilities and is more efficient than GPT models. We choose to apply RoPE embedding and FlashAttention in our model to handle long input sequences. The reason we choose this technique combination rather than other popular ones like LongLoRA (Chen et al., 2023b) is to consider computational cost and algorithmic simplicity. These techniques still keep a one-stage pretraining is simpler and more efficient than two-stage training, which is required by LoRA-based approaches.

**Post-training detection.** With a trained foundation model, we feed a masked transaction into the model and use the reconstruction error as the metric for identifying abnormal transactions. The reason we use the reconstruction error is that the foundation model is trained to learn the patterns of normal transactions. Anomalous transactions tend to have higher reconstruction errors due to their irregularity. Thus, the reconstruction error can be used as an indicator of capturing deviations from learned benign patterns. We also try to build another detection model using the transaction embeddings of the foundation model. As specified in §C.1, we leverage one-class contrastive learning (Sohn et al., 2020) to learn a detection model using either only the <CLS> token embeddings or the embeddings of all tokens. We try to fine-tune the entire model or only train the detection model However, none of these trials can outperform simply masking testing transactions and calculating the reconstruction errors. As such, we stick to the simplest approach, which enables the best detection performance and the least computational cost.

### 4.2 TOKENIZATION

To address these challenges in tokenization we discussed above, `BlockFound` introduces a custom tokenizer specifically designed for the unique multi-model characteristics of blockchain transaction data. We first flatten the raw JSON data into a sequence of function calls and apply a depth-first search to track function callings. We use "[START]" and "[END]" tokens to help the model identify the beginning and end of each function call within the sequence. Additionally, "[Ins]" and "[OUTs]" tokens are used to mark the input and output arguments of functions, which can vary in number. To further distinguish between data types, we use tokens like "data" and "address" to indicate whether the argument is a data value or an address. These special tokens enable the model to clearly recognize the type and boundaries of variable-length information, improving accuracy in transaction tracing.

After pre-processing the transaction trace, we then treat unique hash addresses as individual tokens, which can significantly reduce the overall token count. Given the large number of unique addresses,

```
1. "type": "CALL",
2. "from": "0xc1f351...5d0",
5. "gas": 1962908,
3. "to": "0x4deca5...bac",
4. "func": "0x9fa0bc94",
7. "args": [...],
8. "output": [{"type": "data",
       "data": "0x000000...009"}],
   "calls": [{
        "type": "DELEGATECALL",
        "from": "0x4deca5...bac",
        "gas": 1930278,
10.     "to": "0x35dd16...5e8",
        "func": "0x9fa0bc94",
        "args": [...],
        "output": [...],
        "calls": [...],
        "logs": [...],
   "logMessages": [
9.  "Program PhoeNi... invoke [2]",
    "Program PhoeNi... consumed
            none compute units"],
6.  "value": 0
```

[START]  [CALL]  0xc1f351...5d0  0x4deca5...bac  0x9fa0bc94
start indicator  call indicator  from address  to address  function id
of the calling  1.  2.  3.  4.

0x000000...39c  0x000000...000  [INs]  data  0x476f76...000
5. gas 1962908  6. value 0  7. input  7.
converted to hex  converted to hex  indicator  input type and data

address  0x000000...5d0  data  ......  [OUTs]  data
8. output
indicator

0x000000...009  [logs]  "Program  PhoeNi...units"  [END]
8.  9. log  9.  end indicator
output type and data  indicator  log messages  of the calling

[START]  [DELEGATECALL]  [OOV]  0x35dd16...5e8  0x9fa0bc94
10.  out of
subsequent call's infomation  vocabulary

0x000000...426  [NONE]  [INs]  data  0x476f76...000  ......  [OUTs]
data does not
exist

data  0x000000...009  [END]  [START]  [STATICCALL]  ......  [END]

Figure 1: **Tokenizer of** BlockFound. The figure illustrates how BlockFound tokenizes a transaction by first flattening the nested JSON structure using a depth-first search based on function calls. BlockFound assigns unique tokens to frequently occurring addresses and replaces infrequent addresses with a generic "OOV" token. Special indicator tokens such as "[START]", "[END]", and "[Ins]" are inserted to mark the boundaries of function calls and input/output arguments.

we rank them by frequency and retain the top 7,000 most frequent addresses. Addresses that fall outside the top 7,000 are treated as a single "OOV" token, as shown in Figure 1. In real-world scenarios, frequent addresses are often associated with public smart contracts or exchanges, and preserving them as single tokens improves the system's ability to understand transaction behavior.

For values, as illustrated in Figure 1, there are both decimal numbers (*e.g.,* gas) and hexadecimal numbers (*e.g.,* output data). we first convert all decimal numbers into 40-character hexadecimal format. This approach provides a more compact representation of large numbers as the hexadecimal number is more concise than the decimal number. Small numbers typically begin with "0x000...", which can be learned as a single token. Therefore, this conversion does not lead to a significant increase in token count for small numbers. The consistent formatting of values across all transactions simplifies processing and comprehension for the model.

Unlike hash addresses, log messages often convey information about the same object, such as program status, across different function calls. For example, in Figure 1, log messages like "Program PhoeNi invoke [2]" and "Program PhoeNi consumed none compute units" vary in details but relate to the same event. Treating each log message as a unique token would obscure relationships between messages, which often share common topics. Subword tokenization preserves these connections, ensuring that the model can recognize similarities across different log messages.

The token dictionary size is set at 30,000 to balance the trade-off between token count and information granularity. After allocating space for special tokens and preserved hash address tokens, we apply WordPiece tokenization to learn on the remaining tokens for numbers and log messages.

## 4.3 MODEL DESIGN

BlockFound adapts the RoBERTa model (Liu et al., 2019) to train an auto-encoder specifically for transaction tracing. BlockFound employs a MLM strategy, where $m\%$ of tokens in each transaction are randomly masked. The model is then trained to reconstruct the original transaction from the masked tokens, learning robust representations in the process. However, tokenized transaction data can be significantly longer than typical natural language sequences, posing additional challenges during training. To address these challenges, we incorporate two key techniques: (1) We replace the absolute position embeddings in RoBERTa with Rotary Position Embeddings (RoPE) (Su et al., 2024), which provide more efficient handling of long-range dependencies. (2) We leverage *FlashAttention* (Dao et al., 2022) to accelerate the attention mechanism, improving memory efficiency and reducing computational overhead, making it feasible to train on long transaction sequences.

**Rotary Position Embeddings.** Attention-based models require explicit positional information due to the permutation-invariant nature of the attention mechanism. Traditional approaches such as Sinusoidal(Vaswani, 2017) often struggle with input length constraints. RoPE provides a more flexible solution by rotating the query and key vectors in multi-head attention with a position-dependent rotation matrix. Specifically, given a query $q_i$ and key $k_i$ at position $i$ in a sequence of length $L$, RoPE applies a rotation to each vector as $q_i' = R(i)q_i$ and $k_i' = R(i)k_i$, where $R(i)$ is a sinusoidal function encoding positional information. Unlike absolute embeddings, RoPE introduces relative position dependence, making it more suitable for long-range dependencies and extrapolating to longer sequences. This enables models using RoPE to effectively train on long transaction sequences.

**FlashAttention** is a memory-efficient algorithm designed to compute exact attention while optimizing both time and memory usage. The key innovation lies in addressing a bottleneck in standard attention mechanisms, where frequent data transfer between fast on-chip GPU memory (SRAM) and slower high-bandwidth memory (HBM) leads to inefficiencies. FlashAttention mitigates this by splitting the Query/Key/Value matrices into smaller blocks and processing them incrementally, reducing the need for frequent data movement to and from HBM. Additionally, in the backward pass, it recomputes large intermediate results such as attention scores, trading extra computation for fewer memory operations. This approach significantly reduces memory overhead and speeds up attention computation without compromising model accuracy, making it well-suited for handling long sequences in resource-constrained environments.

### 4.4 POST-TRAINING DETECTION

After training, we can deploy `BlockFound` for detecting anomalous transaction sequences. The motivation behind applying `BlockFound` for transaction anomaly detection is that since the model is trained on benign transaction sequences, it can accurately predict masked tokens if the sequence is also benign. Hence, the anomalous score of a transaction can be derived based on the prediction results on the masked tokens. Specifically, for a given transaction, we randomly mask a ratio of the tokens, similar to the training process, and input the masked sequence into the trained model. The probability distribution over the possible tokens for each MASK position represents the likelihood of each token in that position. We construct a candidate set of the top-$s$ most likely tokens for each masked position. If the true token appears within the top-$s$ candidate set, we consider the token as benign. Conversely, if the true token is not in the top-$s$ candidate set, it is treated as anomalous. The reason why we do not directly predict based on the most likely token is that the addresses and values are more challenging than nature language texts to predict, and having a candidate set tolerant to the prediction error is more reasonable. After ranking the transactions by the anomalous score, we can select the top $k$ transactions with the highest anomalous score as anomalous. $k$ can be dynamically adjusted based on how the smart contract developers trade off between false positives and security of the transactions.

## 5 EXPERIMENTS

In this section, we present the experimental evaluation of `BlockFound` in anomalous transaction detection. We begin by introducing the experimental setup, including the dataset and evaluation metrics. Then, we compare `BlockFound` to other detection methods to showcase the effectiveness of `BlockFound`. Additionally, we conduct ablation studies to analyze the impact of hyper-parameters of `BlockFound`.

### 5.1 EXPERIMENTAL SETUP

**Dataset.** We primarily focus on Ethereum and Solana transactions in our experiments. We sample transactions from interactions with 5 DeFi applications for Ethereum and 10 applications for Solana to ensure diverse transaction patterns. For each DeFi application, transactions are ordered by their block timestamps and split into 80% for training and 20% for evaluation as benign transactions. This per-application sequential split is crucial to prevent time travel data leakage, ensuring that the model is trained exclusively on past data without access to future information. Such a methodology can maintain the integrity of performance metrics by avoiding artificially inflated results that could arise if the model inadvertently learned from future transactions.

Specifically, our Ethereum dataset consists of 3,383 benign transactions for training, 709 benign transactions for testing, and 10 malicious transactions. The data was collected from October 2020 to April 2023. For Solana, our training dataset comprises 35,115 transactions, while the testing dataset includes 1,500 benign transactions and 18 malicious transactions. The Solana data is sampled in September 2023 and December 2023, spanning a two-month period due to the availability of transaction data. The benign transactions for both Ethereum and Solana were sampled and manually cleaned to remove transactions unrelated to the target applications or failed transactions. The malicious transactions were sourced from verified transaction vendors, including Zengo, TRM Labs, and CertiK, and manually verified to ensure their malicious nature. Note that the malicious transactions are also sampled from these selected DeFi applications. To mitigate the risk of data leakage, we ensured that the malicious transactions occurred after the sampling periods of benign transactions. This approach guarantees that the model is trained solely on known benign transactions up to the cutoff dates, preventing any inadvertent exposure to future anomalous patterns during training.

**Evaluation Metrics.** We adopt the evaluation methodology from BlockGPT (Gai et al., 2023), where transactions are ranked based on their detection scores produced by the models. Specifically, the top-$k$ transactions with the highest scores are labeled as anomalous, while the remaining transactions are classified as benign. The binary classification performance is evaluated using the following metrics: *False Positive Rate* (FPR), *Recall*, and *Precision*. In our experiments, we select $k$ values from the set 5, 10, 15 for Ethereum and 10, 15, 20 for Solana considering the number of collected to evaluate the model's performance at different detection thresholds. A larger $k$ value indicates a higher detection threshold, potentially leading to more false positives but could detect more anomalous transactions, which can be varied based on how the DeFi owner wants to trade off between false positives and security.

**Model architecture and hyper-parameters.** We use the BERT-base architecture, which includes 100 million parameters, for training the Ethereum task, and the BERT-large architecture, with 300 million parameters, for training the Solana task. We set the learning rate to 5e-5 and use a batch size of 32 for the Ethereum task and 4 for the Solana task, respectively. For the Ethereum task, the maximum sequence length is set to 1,024 tokens, while for the Solana task, we increase the maximum sequence length to 8,192 tokens to accommodate the longer transactions. Please refer to §C.1 for a detailed setup of the training hyper-parameters for both datasets. In the inference phase, we set the mask ratio $g$ to 15% and the number of candidate tokens $s$ is set to 3 on both datasets.

**Baselines.** To evaluate the effectiveness of `BlockFound`, we compare it against several anomalous transaction detection methods: ❶ **BlockGPT** (Gai et al., 2023): It pretrains a causal transformer model on the transaction corpus to learn typical benign transaction patterns. The underlying intuition is that anomalous transactions deviate from these learned patterns and are therefore difficult to predict. BlockGPT calculates the sum of the conditional log-likelihoods for each token in a transaction sequence, with lower likelihoods indicating potential anomaly. The top-$k$ transactions with the lowest scores are flagged as anomalous. ❷ **Doc2Vec** (Le & Mikolov, 2014): It represents the transaction as a bag of words and leverages the distributed representation of words to represent the transaction. These vectorized transactions are then analyzed using a GMM to estimate the probability of each transaction being anomalous. This probabilistic approach allows for the identification of anomalous transactions based on their likelihood within the learned distribution. ❸ **GPT-4o**: This method utilizes a state-of-the-art commercial language model to assign an anomaly score ranging from 0 to 100 to each transaction. This approach relies on the extensive pre-training of the language model, which could potentially encompass a wide variety of anomalous transaction patterns, enabling it to detect suspicious activities based on learned knowledge. ❹ **Heuristic-based methods**: Previous study (Risse & Böhme, 2024) has highlighted that machine learning models can sometimes achieve decent detection rates by leveraging trivial features like input length in detection tasks. To explore this, our heuristic-based approach uses the length of a transaction as the sole feature, operating under the assumption that anomalous transactions are typically longer than benign ones.

By comparing `BlockFound` with these diverse baselines, we aim to demonstrate its superior performance in accurately identifying anomalous transactions while mitigating the impact of potential confounding factors present in other detection methods.

| Method | k=10 | | | k=15 | | | k=20 | | |
|---|---|---|---|---|---|---|---|---|---|
| | FPR | Recall | Precision | FPR | Recall | Precision | FPR | Recall | Precision |
| **BlockGPT** | 0.47% | 16.67% | 30% | 0.73% | 22.22% | 26.67% | 1% | 27.78% | 25% |
| **Doc2Vec** | 0.67% | 0% | 0% | 1% | 0% | 0% | 1.13% | 0% | 0% |
| **GPT-4o** | 0.67% | 0% | 0% | 1% | 0% | 0% | 1.13% | 0% | 0% |
| **Heuristic** | 0.67% | 0% | 0% | 1% | 0% | 0% | 1.13% | 0% | 0% |
| BlockFound | **0.13%** | **44.44%** | **80%** | **0.2%** | **66.67%** | **80%** | **0.47%** | **72.22%** | **65%** |

Table 1: **Performance comparison with different $k$ values for Solana.**

| Method | k=5 | | | k=10 | | | k=15 | | |
|---|---|---|---|---|---|---|---|---|---|
| | FPR | Recall | Precision | FPR | Recall | Precision | FPR | Recall | Precision |
| **BlockGPT** | 0.14% | 40% | 80% | 0.42% | 70% | 70% | 0.99% | 80% | 53.33% |
| **Doc2Vec** | 0.56% | 10% | 16.67% | 1.12% | 20% | 18.18% | 1.83% | 20% | 12.5% |
| **GPT-4o** | 0.28% | 30% | 37.5% | 0.98% | 30% | 23% | 1.55% | 40% | 21% |
| **Heuristic** | 0.14% | 40% | 80% | 0.42% | 70% | 70% | 1.13% | 70% | 46.67% |
| BlockFound | **0%** | **50%** | **100%** | **0.28%** | **80%** | **80%** | **0.97%** | **80%** | **53.33%** |

Table 2: **Performance comparison with different $k$ values for Ethereum.**

## 5.2 EXPERIMENTAL RESULTS

**Comparison with Baselines.** We show the FPR, Recall, and Precision of `BlockFound` and other baselines in Table 1 and Table 2. As the results show, `BlockFound` outperforms all baseline methods across various $k$ values for both the Ethereum and Solana datasets. Notably, on the Solana dataset, most baseline methods (Doc2Vec, GPT-4o, and Heuristic) consistently fail to detect any anomalous transactions, achieving a recall and precision of 0% for all $k$ values. This indicates that all transactions flagged as anomalous by these methods are, in fact, benign. While BlockGPT is able to detect some anomalous transactions, its recall and precision are significantly lower than those of `BlockFound`. For instance, at $k = 20$, BlockGPT achieves only a 27.78% recall with a FPR of 1%. In contrast, `BlockFound` detects the majority of anomalous transactions (*i.e.,* 13 out of 18) with a lower FPR of 0.47%.

We have the following potential reasons for the failure of these baseline methods: 1) Doc2Vec's approach of representing transactions as a bag of words likely fails due to its inability to capture the sequential dependencies and contextual nuances crucial for distinguishing between benign and anomalous transactions. 2) Despite its extensive pre-training, GPT-4o may underperform because it is not specifically fine-tuned for blockchain-specific anomalous transaction detection, making it less effective in identifying such domain-specific anomalies. 3) The heuristic method fails when the heuristics are not accurate for those anomalous transactions that have similar length as benign transactions. 4) BlockGPT, which shares the most similar idea with `BlockFound`, fails to detect anomalous transactions because the casual language model structure may not be suitable for detection task. For each token in the transaction, it only considers preceding information while `BlockFound` can analyze both previous and subsequent information for tokens to predict.

In contrast to baseline methods's failure, `BlockFound` demonstrates strong performance with significantly lower FPRs and much higher recall and precision scores, especially as the $k$ threshold increases. For example, at $k = 10$ on the Ethereum dataset, `BlockFound` achieves an FPR of 0.28%, a recall of 80%, and a precision of 80%, which means `BlockFound` can successfully detect 8 out of 10 anomalous transactions while only predicting 2 false positives.

These results highlight the effectiveness of `BlockFound` in accurately identifying anomalous transactions while minimizing false positives, thereby demonstrating its superiority over existing detection methods in both Ethereum and Solana environments. The baselines' failure to detect anomalous transactions also underscores the challenge of this task and the importance of leveraging advanced methods like `BlockFound` for robust blockchain transaction security.

**Effect of Core Components.** We conduct an ablation study on `BlockFound` by removing each core component individually to analyze its impact on detection performance with the Solana dataset.

The first component we ablate is the *tokenizer*, which is specifically designed to handle transaction data in `BlockFound`. To evaluate its significance, we replace the custom tokenizer with the default WordPiece tokenizer from BERT, allowing us to observe how much this tailored tokenization contributes to the model's success. Next, we examine the effect of *log message*, which is the printed information when executing these transactions. As mentioned in §4.2, we use subword tokenization to encode the log messages in order to preserve their context information. Here, we substitute this approach by treating each log message as a unique token, similar to how we treat the hash addresses, and measure the resulting change in performance. Lastly, we study the effect of the *RoPE embedding*, which we employ to capture the relative position information between tokens. In this ablation, we replace it with the default absolute positional embedding.

The results are presented in the upper half of Table 3. From these experiments, we draw the following conclusions. First, substituting our customized tokenizer with the default BERT tokenizer, while keeping all other components unchanged, caused the model to fail to detect any anomalous transactions (*i.e.,* the recall was 0 across different $k$ values). This underscores the importance of our customized tokenizer, as the default BERT tokenizer, trained on general text data, is unable to capture the complex structure of specific transaction traces. Second, we observed that the model also struggled to differentiate between benign and anomalous transactions when we altered the log message encoding strategy. This suggests that the log messages may contain key information about the transaction status in the Solona task, and an appropriate encoding method, such as a subword tokenizer, can extract this information effectively. Lastly, replacing our relative positional embeddings with absolute positional embeddings led to a significant drop in model performance, with a decrease in recall of nearly 20% to 30% across various $k$ values. This emphasizes the importance of relative positional embeddings for effectively handling long sequences (*e.g.,* a sequence length of 8192).

We also conduct an ablation study on the impact of FlashAttention on the training time and memory usage of `BlockFound` in Table 9. The results show that the integration of FlashAttention reduces the training time and memory usage while maintaining the detection performance. Furthermore, we investigate the selection of the base model in Table 10 by replacing the RoBERTa model with other state-of-the-art BERT-like models like DeBERTa (He et al., 2020). The results demonstrate that `BlockFound` achieves consistent performance and is agnostic to the choice of base model.

**Hyper-parameters sensitivity analysis.** We further investigate the impact of key hyper-parameters and model architecture on the final model performance in the Solana task. Specifically, we introduce two additional hyper-parameters during detection phrase: the detection mask percentage $g$ and the number of candidate tokens $k$ used when calculating the mask prediction accuracy. By varying $g$ and $s$ within {5, 10, 15} and {1, 3, 5}, respectively, we assess the model's robustness to these parameters. Additionally, While BERT-large is the default model on Solana dataset, we replace it with BERT-base to evaluate the influence of different model architectures on the final performance.

As shown in the lower half of Table 3, our model demonstrates a degree of robustness to variations in $g$ and $s$. Recall that the default values for $g$ and $s$ in `BlockFound` are 15% and 3, respectively. Notably, `BlockFound`-s=1 even outperforms `BlockFound` when $k = 20$, suggesting that a simpler set of hyperparameters can still achieve relatively good performance. However, when the model architecture is switched from BERT-large to BERT-base, a noticeable performance drop occurs on the Solona dataset. This is likely due to the dataset's large number of training samples (*i.e.,* almost 30,000) and longer sequence length (*i.e.,* 8,192 tokens), which smaller models like BERT-base struggle to handle effectively.

## 6  DISCUSSION

**Dataset Size and Quality.** In our evaluation, we use a dataset of 28 malicious transactions, which represent real-world exploits of smart contract vulnerabilities in the selected DeFi applications. Collecting a larger set of verified malicious transactions is non-trivial due to the manual effort required for verification and the need to use only publicly available data to maintain privacy standards and enable open-sourcing. Prior work Gai et al. (2023) identified a total of 116 malicious transactions; however, they were unable to share these transactions or their sources with us due to privacy concerns. We have open-sourced our datasets and models to foster further research and expansion of the malicious transaction dataset in the future.

| Models | k=10 | | | k=15 | | | k=20 | | |
|---|---|---|---|---|---|---|---|---|---|
| | **FPR** | **Recall** | **Precision** | **FPR** | **Recall** | **Precision** | **FPR** | **Recall** | **Precision** |
| `BlockFound` | **0.13%** | **44.44%** | **80%** | **0.2%** | **66.67%** | **80%** | 0.47% | 72.22% | 65% |
| - Tokenizer | 0.67% | 0% | 0% | 1% | 0% | 0% | 1.33% | 0% | 0% |
| - Log message | 0.67% | 0% | 0% | 1% | 0% | 0% | 1.33% | 0% | 0% |
| - RoPE | 0.4% | 22.22% | 40% | 0.53% | 38.89% | 46.67% | 0.80% | 44.44% | 40% |
| `BlockFound-100m` | 0.6% | 5.56% | 10% | 0.93% | 5.56% | 6.67% | 1.27% | 5.56% | 5% |
| `BlockFound-g=10` | 0.27% | 33.33% | 60% | 0.4% | 50% | 60% | 0.53% | 66.67% | 60% |
| `BlockFound-g=15` | 0.27% | 33.33% | 60% | 0.4% | 50% | 60% | 0.53% | 66.67% | 60% |
| `BlockFound-s=1` | 0.13% | 44.44% | 80% | 0.27% | 61.11% | 73.33% | **0.4%** | **77.78%** | **70%** |
| `BlockFound-s=5` | 0.13% | 44.44% | 80% | 0.27% | 61.11% | 73.33% | 0.47% | 72.22% | 65% |

Table 3: **Ablation study on** `BlockFound` **for Solana.**

**Tokenizer and Model Adaptability.** In our approach, we initially build the tokenizer using a large transaction corpus. To maintain optimal detection performance, we recommend periodically rebuilding the tokenizer to keep it aligned with current transaction patterns. Additionally, our model's training on benign data allows it to learn typical transaction patterns and detect anomalies based on deviations from these patterns, providing a level of adaptability to new or previously unseen malicious strategies. However, if new attack strategies closely mimic benign patterns, detection may be challenging. To address this, we recommend periodically retraining the model on updated data to ensure optimal detection accuracy as the blockchain ecosystem evolves.

**Fine-Tuning GPT-4o.** As shown in §5.2, directly using GPT-4o as a detector results in poor performance. We further fine-tune GPT-4o via OpenAI API on Ethereum dataset but still observe limited performance. We hypothesize that this is because the fine-tuning API is coarse-grained and does not allow next token prediction nor customization of tokenization, which is crucial for our task. Detailed results are shown in Table 7.

**Robustness to Noise.** In blockchain transactions, the ratio of benign to malicious transactions is typically highly imbalanced. Give this high imbalance, even if some potential malicious samples are inadvertently included in the training set, their impact is minimal, as the model predominantly learns the representation of the majority class (benign transactions). To assess the model's robustness to noise, we conduct an experiment simulating an extreme case where half of the malicious transactions were intentionally included in the training set for the Ethereum dataset. While this scenario caused a slight drop in detection performance, the model remained effective. These results demonstrate that our approach maintains robustness to a reasonable level of noise in the data.

**Future Work.** Our work opens up several avenues for future research. First, explainable AI is critical for deploying any AI system in production, particularly within the financial sector of DeFi. Integrating explanation tools for LLM can enhance the transparency of `BlockFound` and will be essential to better understand the patterns it learns and to identify and mitigate potential biases in its predictions. Second, although our experiments show that directly using GPT-4o or fine-tuned GPT-4o as a detector results in poor performance, we believe that more sophisticated approaches, such as advanced prompt engineering and integrating on-chain tools (*e.g.,* verifying addresses on the blockchain), could significantly improve the performance of LLM-based detectors. Lastly, we plan to extend our evaluation to additional blockchain platforms such as Binance Smart Chain and Polkadot, which differ in consensus mechanisms and transaction patterns, to further validate the adaptability of our approach. We leave these explorations for future work.

## 7 CONCLUSION

In this work, we presented `BlockFound`, a transformer-based model designed for detecting anomalous transactions in DeFi ecosystems such as Ethereum and Solana. By leveraging masked language modeling and carefully designed tokenization techniques, `BlockFound` efficiently handles the complexity and diversity of transaction data. Additionally, we open-sourced the code, model, and datasets used in this work, making `BlockFound` the first open-source solution for LLM-based anomalous transaction detection in DeFi. We hope that this contribution will serve as a valuable resource for the research community, facilitating further advancements in the development of scalable and robust anomalous transaction detection systems.

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

# A  PSEUDO ALGORITHM OF `BlockFound`

We present the pseudo algorithm of `BlockFound` in Algorithm 1 to help readers understand the workflow of `BlockFound`.

---

**Algorithm 1: Workflow of `BlockFound`**

---

**Input:** Benign transactions $\mathcal{D} = \{D_1, \ldots, D_N\}$, transactions to be predicted $\mathcal{P} = \{T_1, T_2, \ldots, T_M\}$, mask percentage $m$, detection mask percentage $g$, top-$s$ candidates, threshold $k$

1 **Tokenization:**
2 Initialize tokenizer $\mathcal{T}$ with preserved address tokens and special tokens
3 Train subword tokenization on remaining data to generate the final token dictionary
4 Save tokenizer $\mathcal{T}$
5 **Training Phase:**
6 **for** *each transaction $D_i \in \mathcal{D}$* **do**
7 $\quad$ Tokenize $D_i$ using $\mathcal{T}$
8 $\quad$ Randomly select $m$ tokens from $D_i$ and mask them: $D_i' = \text{Mask}(D_i, m)$
9 $\quad$ Train the model $\mathcal{M}$ to minimize the loss function: $\mathcal{L} = \sum_{i=1}^{N} \mathbb{E}_{D_i} [\log P(D_i | D_i', \mathcal{M})]$
10 **end**
11 Save the trained model $\mathcal{M}^*$
12 **Detection Phase:**
13 **for** *each transaction $T_j \in \mathcal{P}$* **do**
14 $\quad$ Tokenize $T_j$ using $\mathcal{T}$
15 $\quad$ Randomly select $g\%$ of tokens and mask them: $T_j' = \text{Mask}(T_j, g)$
16 $\quad$ Use the trained model $\mathcal{M}^*$ to predict the top-$s$ tokens for each masked token position:
$$\hat{T}_j = \{\hat{t}_{i,1}, \hat{t}_{i,2}, \ldots, \hat{t}_{i,s} \text{ for } i = 1, \ldots, n\}$$
$\quad$ Calculate the failed prediction ratio (abnormality score) for $T_j$:
$$\text{Score}(T_j) = \frac{1}{n} \sum_{i=1}^{n} \mathbb{I}(t_i \notin \{\hat{t}_{i,1}, \ldots, \hat{t}_{i,s}\})$$
17 **end**
18 **return** *Top-k transactions $\hat{\mathcal{P}}$ ranked by anomaly score $\text{Score}(T_j)$*

---

# B  ADDITIONAL DETAILS ON DATASET

Here we provide additional details on the dataset used in our experiments.

**Address Frequency.** To balance training efficiency and information retention, we rank all unique addresses in the dataset by frequency and retain only the top 7,000 addresses for training. For Ethereum, this covers the majority of unique addresses, as there are only 7,335 addresses in total, with the remaining addresses appearing just once in the training set. For Solana, the dataset contains 56,203 unique addresses, and retaining all of them would significantly increase the embedding size, making training computationally infeasible due to the high resource demands. Notably, the addresses excluded from the top 7,000 in Solana appear fewer than 10 times in the training set, contributing minimally to the overall information.

High-frequency addresses typically correspond to smart contracts, token addresses, or other entities that are frequently accessed and more significant for classification tasks. Conversely, low-frequency addresses, such as individual user wallets, often carry less relevance for anomaly detection. Including these low-frequency addresses would increase model complexity and training time without yielding significant performance gains. By focusing on the most frequent 7,000 addresses, we ensure a practical trade-off between training efficiency and the retention of critical information for effective anomaly detection.

**Potential Duplication in Transaction Data.** Contract templates are wildly used in real-world smart contract development, leading to different smart contracts may offering similar or even identical APIs to the users. This could cause potential duplication in the transaction data. To assess the

extent of this issue, we conduct a 5-gram BLEU similarity analysis on our dataset, choosing 5-gram to avoid false positives caused by indicator tokens such as "[START]" and "[CALL]." Our analysis reveal that only 0.05% of transaction pairs in the Ethereum dataset exhibit a BLEU similarity exceeding 0.7, with 0.023% surpassing 0.8. These highly similar transactions may indeed result from the use of contract templates. Given the low similarity ratio in our data, we do not consider potential duplication a significant issue.

## C    DETAILED EXPERIMENTAL RESULTS

### C.1    IMPLEMENTATION DETAILS

**Our method**    We detail the hyper-parameters and training process of our customized language models, each trained from scratch for either the Solana or Ethereum tasks. Recall that for the Solana dataset, the model is based on a BERT-large architecture, with a hidden dimension of 1024, 24 hidden layers, and 16 attention heads. For the Ethereum dataset, the model uses a BERT-base architecture, with a hidden dimension of 768, 12 hidden layers, and 12 attention heads. The complete set of training hyper-parameters is detailed in Table 4 and Table 5. The Solana model was trained over two days using eight A100 GPUs, while the Ethereum model required around 2 hours of training on the same hardware.

| config | value |
| --- | --- |
| optimizer | Adam (Kingma, 2014) |
| base learning rate | 5e-5 |
| weight decay | 0.0 |
| gradient accumulation step | 10 |
| optimizer momentum | $\beta_1, \beta_2 = 0.9, 0.999$ |
| batch size | 3 |
| learning rate schedule | cosine decay |
| warmup epochs | 1 |
| total epochs | 10 |
| max sequence length | 8192 |

Table 4: **Configuration of training setup on Solana dataset.**

| config | value |
| --- | --- |
| optimizer | Adam |
| base learning rate | 5e-5 |
| weight decay | 0.0 |
| gradient accumulation step | 10 |
| optimizer momentum | $\beta_1, \beta_2 = 0.9, 0.999$ |
| batch size | 20 |
| learning rate schedule | cosine decay |
| warmup epochs | 10 |
| total epochs | 100 |
| max sequence length | 1024 |

Table 5: **Configuration of training setup on Ethereum dataset.**

**Baselines**    We employ four baseline methods: BlockGPT, Doc2Vec, GPT-4o, and Heuristic. For BlockGPT, as the source code was unavailable, we contact the author and implement BlockGPT based on their guidance. For the Doc2Vec approach, as described by Gai et al. (2023), we first apply Doc2Vec (Le & Mikolov, 2014) to extract features from the pre-processed and flattened traces of training transactions, as is shown in Figure 1. After obtaining the feature representations, we build a GMM to model the training transactions' distribution using the Sklearn library (Pedregosa et al., 2011) with default hyper-parameters. During evaluation, for each transaction, we extract its feature

using Doc2Vec and computed its anomaly score as the negative log-likelihood under the GMM. For the heuristic method, the anomalous score of a given transaction is determined by the sequence length of the corresponding flattened traces, with longer traces indicating a higher probability of anomaly behavior. For GPT-4o, we use the above prompts to instruct the LLM to give a score between 0 and 100. We use chain-of-thought (COT) (Wei et al., 2022) prompting to further improve the performance of GPT-4o. Additionally, we integrate human prior knowledge into the LLM by providing it with the list of known anomalous patterns to help it make more accurate predictions.

---

**Prompt for GPT-4o method**

You are a blockchain security expert tasked with determining whether a given blockchain transaction is anomalous. Please evaluate the transaction step by step and consider the following aspects:
1. Analyze the sender and recipient addresses to check if they have been involved in known anomalous activity.
2. Assess the transaction value and fee to identify any unusual patterns that might indicate suspicious behavior.
3. Examine the transaction's input data, including any smart contract interactions, to see if they match known attack vectors.
4. Consider the timing and frequency of the transaction relative to previous transactions from the same address.
Assign a score between 0 and 100, where 0 means completely benign and 100 means highly anomalous. Provide a clear explanation of the reasoning behind your score. Finally, return the result in the following JSON format:
#json
{ "reason": "Detailed explanation of why the transaction is considered anomalous or benign.",
"score": "A number between 0 and 100 representing the likelihood of the transaction being anomalous."
}
Transaction details: [Insert transaction data here]

---

### C.2 ADDITIONAL EXPERIMENTS

**Post-Detection Methods.** As mentioned in §4.1, we also explore post-detection methods using a one-class contrastive learning approach. In this experiment, we apply the method to the Ethereum dataset. Specifically, after pre-training our customized LLM on the Ethereum task, we extract feature representations for each transaction by either using the <CLS> token embeddings or the average embeddings of all tokens. We then perform one-class contrastive learning on the training set, treating positive samples as those originating from the same DeFi application and negative samples as those from different DeFi applications. Through this contrastive learning process, we aim to obtain more robust feature representations of the transactions. Finally, we apply kernel density estimation (KDE) to the features learned through one-class contrastive learning, where a lower density score for a transaction indicates a higher probability of it being anomalous. Details of the hyper-parameter settings can be found at `https://shorturl.at/9dFL1`.

As shown in Table 6, neither <CLS>-CL (*i.e.,* one-class contrastive learning using input feature from <CLS> token embeddings) nor Average-CL (*i.e.,* using input feature from the average embeddings of all tokens) outperforms our method. Compared with post-detection using one-class contrastive learning method, `BlockFound` achieves relatively good performance without requiring additional computation resources. Therefore, we continue to use the simplest approach—our current masked prediction method—as the post-detection method.

**Fine-Tuning GPT-4o.** We fine-tune GPT-4 (version 2024-08-06) using the Ethereum dataset to evaluate its performance on domain-specific tasks following **BlockGPT**'s approach. However, our method deviated from traditional token-by-token iteration approaches due to the limitations of OpenAI's fine-tuning API, which supports only instruction-response style fine-tuning. Instead, we divide each benign transaction into two halves: the first half served as the input, and the second half as the

| Method | k=5 | | | k=10 | | | k=15 | | |
|---|---|---|---|---|---|---|---|---|---|
| | FPR | Recall | Precision | FPR | Recall | Precision | FPR | Recall | Precision |
| <CLS>-CL | 0.28% | 30% | 60% | 0.28% | 80% | 80% | **0.85%** | **90%** | **60%** |
| Average-CL | 0.14% | 40% | 80% | 0.28% | 80% | 80% | 0.97% | 80% | 53.33% |
| BlockFound | **0%** | **50%** | **100%** | **0.28%** | **80%** | **80%** | 0.97% | 80% | 53.33% |

Table 6: **Performance comparison of different post-detection methods for Ethereum.**

target response. This method aimed to enable the model to predict transaction details. The error between the predicted and actual transaction is used as the anomaly score. We summarize the results in Table 7.

The results indicate no significant improvement over the default GPT-4o model. Several factors may contribute to this outcome:

- **Training Budget Constraints**: Our fine-tuning costs are approximately $950, limiting the number of training iterations.
- **Coarse-Grained Approach**: The half-half prediction strategy may not have captured the intricate details of transaction patterns.
- **Tokenization Challenges**: GPT-4o's default tokenization struggles with specific data types, such as blockchain addresses and numerical patterns, reducing its ability to learn precise representations.

To overcome these limitations, future efforts could include:

- Developing more fine-grained fine-tuning strategies.
- Exploring additional tools to preprocess blockchain-specific inputs, such as addresses and numbers.
- Leveraging models with customizable tokenization and greater control over training objectives.

| Model | k=5 | | | k=10 | | | k=15 | | |
|---|---|---|---|---|---|---|---|---|---|
| | FPR | Recall | Precision | FPR | Recall | Precision | FPR | Recall | Precision |
| GPT-4o | 0.28% | 30% | 37.5% | 0.98% | 30% | 23% | 1.55% | 40% | 21% |
| GPT-4o-FT | 0.28% | 30% | 37.5% | 0.98% | 30% | 23% | 1.55% | 40% | 21% |

Table 7: **Performance comparison of fine-tuned GPT-4o and GPT-4 on Ethereum for various $k$ values.**

**Robustness to Noise.** We intentionally modified the training data to include 50% of the malicious transactions while keeping the rest of the data unchanged for the Ethereum dataset. As shown in Table 8, the detection performance of BlockFound is still relatively good, achieving a recall of 60% for a detection threshold $k = 10$. These results demonstrate that our approach maintains robustness to a reasonable level of noise and inaccurate information in the data.

| Model | k=5 | | | k=10 | | | k=15 | | |
|---|---|---|---|---|---|---|---|---|---|
| | FPR | Recall | Precision | FPR | Recall | Precision | FPR | Recall | Precision |
| No Noise | **0%** | **50%** | **100%** | **0.28%** | **80%** | **80%** | **0.97%** | **80%** | **53.33%** |
| With Noise | 0.14% | 40% | 80% | 0.56% | 60% | 60% | 1.26% | 60% | 40% |

Table 8: **Performance comparison of models with and without noise for Ethereum for various $k$ values.**

## C.3 Ablation Study

**Impact of FlashAttention.**    To evaluate the impact of FlashAttention on the training efficiency and resource utilization of `BlockFound`, we conduct an ablation study on Ethereum and Solana datasets. The results are shown in Table 9.

The integration of FlashAttention significantly improves training efficiency by optimizing attention computation. For Ethereum, FlashAttention reduces the running time from 9,415 seconds to approximately 7,000 seconds, as shown in the table. Additionally, it nearly halves the GPU memory usage, enabling more efficient use of hardware resources.

For Solana, the impact of FlashAttention is even more pronounced. Without FlashAttention, the model cannot handle even a batch size of 1 on an 80GB A100 GPU due to memory constraints. With FlashAttention, the training process becomes feasible, allowing a batch size of 2 while maintaining memory efficiency.

These results highlight the critical role of FlashAttention in handling long sequences and enabling scalable training for large datasets without sacrificing detection performance. Also, enabling FlashAttention has no noticeable impact on the accuracy of the model for Ethereum, as it achieves the same detection performance as the model without FlashAttention. This aligns with its design goal of optimizing computational efficiency rather than altering model representations or outputs. This study demonstrates that FlashAttention is essential for enabling efficient training on long sequences and large datasets while maintaining detection performance.

| Dataset | Training Time (s) | | GPU Memory Usage (GB) | |
|---|---|---|---|---|
| | With FlashAttention | Without FlashAttention | With FlashAttention | Without FlashAttention |
| **Ethereum** | 7,042 | 9,415 | 41.5 | 78.4 |
| **Solana** | 170,210 | - | 79.4 | - |

Table 9: **Impact of FlashAttention on training time and GPU memory usage for Ethereum and Solana datasets.**

**Impact of Base Model.**    To ensure that the choice of the base model does not significantly influence the performance of our framework, we conducted additional experiments with alternative state-of-the-art BERT-like models, such as DeBERTa (He et al., 2020), on the Ethereum dataset. These models are selected for their outstanding ability in NLP tasks. As shown in Table 10, the results achieved with DeBERTa are consistent with those of RoBERTa. This validation confirms that our framework is agnostic to the specific choice of base model, offering flexibility in adapting to other transformer-based architectures. Future work may explore additional models to further generalize the framework's applicability.

| Model | k=5 | | | k=10 | | | k=15 | | |
|---|---|---|---|---|---|---|---|---|---|
| | FPR | Recall | Precision | FPR | Recall | Precision | FPR | Recall | Precision |
| **DeBERTa** | 0% | 50% | 100% | 0.28% | 80% | 80% | 0.97% | 80% | 53.33% |
| **RoBERTa** | 0% | 50% | 100% | 0.28% | 80% | 80% | 0.97% | 80% | 53.33% |

Table 10: **Performance comparison of different base models on Ethereum for various $k$ values.**

