# OpenReview forum: "BlockFound: Customized blockchain foundation model for anomaly detection"
_ICLR.cc/2025/Conference — Submitted to ICLR 2025_

### Official Review · Reviewer_NnUh · 2024-10-22

**Soundness:** 2
**Presentation:** 3
**Contribution:** 3
**Rating:** 6
**Confidence:** 4

**Summary:**

This paper introduces BlockFound, a customized foundation model designed for detecting anomalous blockchain transactions. It modifies BERT model by using RoPE embedding and FlashAttention to suit the characteristics of blockchain transactions. The performance of BlockFound is evaluated using real Ethereum and Solana transaction data, demonstrating its superiority on key performance metrics over baseline methods.

**Strengths:**

First, the idea of using a LLM to detect anomalous transactions in blockchain is innovative.
Second, the authors have appropriately adapted the LLM for blockchain transaction data, and these modifications have been proven effective through ablation study.
Third, the authors evaluated BlockFound using actual blockchain data, and the results demonstrate that BlockFound achieves better performance compared to baseline methods.

**Weaknesses:**

1. The introduction of this paper needs to sufficiently explain the motivation behind utilizing an LLM for detecting anomalous transactions in blockchain environments. It lacks evidence to support the claim that blockchain attacks are mainly due to anomalous transactions. Indeed, existing works have discussed that vulnerabilities in consensus protocols can lead to attacks authors mentioned, including double-spending attacks, even when transactions are valid and normal. From another perspective, even if anomalous transactions are detected, the paper does not provide evidence demonstrating that such detection can effectively prevent attacks. Additional references or evidence should be provided to justify the need for anomaly detection in blockchain transactions.
2. The paper needs to clearly define what is an anomalous transaction. The ground truth for such anomalies is not clear. Only with a known ground truth of anomalous transactions can we discuss whether "using the reconstruction errors as the metric for anomaly detection" is appropriate.
3. There is a need for more detailed analysis on the reliability of the datasets used (types, the number of training samples, the number of testing samples, etc.). For example, the statement "our Ethereum dataset consists of 3,383 benign transactions for training, 709 benign transactions for testing, and 10 malicious transactions. The data was collected from October 2020 to April 2023." raises questions. Does this imply there were only ten malicious transactions on Ethereum from October 2020 to April 2023, or were these ten selected by the authors? If selected, what were the criteria for their selection?
4. Why did the study use two different model architectures for different datasets? Does this indicate that the learning models for malicious transaction patterns lack transferability, necessitating the independent training of a model for each specific blockchain/dataset?
5. While the paper is clear and logically presented, it is more like a technical report than an academic paper.

**Questions:**

Please see weaknesses.

---

> ### Author Response · Authors · 2024-11-19
>
> > **Reviewer's Comment:** The introduction of this paper needs to sufficiently explain the motivation behind utilizing an LLM for detecting anomalous transactions in blockchain environments. ......Additional references or evidence should be provided to justify the need for anomaly detection in blockchain transactions.
>
> **Response:** Thank you for your insightful comments. While we acknowledge that consensus protocol vulnerabilities have been studied extensively in the academic literature, we note that such vulnerabilities are often very challenging to exploit in real-world blockchain environments. To the best of our knowledge, there are few, if any, practical instances of attacks exploiting consensus-level vulnerabilities in deployed systems in recent years.
>
> In contrast, most real-world attacks target logic or implementation flaws within smart contracts, which can manifest as anomalous transaction behaviors. Therefore, our study focuses on detecting anomalies at the transaction level to identify these types of vulnerabilities. By detecting such anomalies, our approach enables transaction emergency stop, which can help prevent losses when malicious behavior is detected. Prior works also highlight the importance of anomaly detection in minimizing the loss and upholding the integrity[1,2,3].
>
> We have incorporated additional references in the revised paper to further support this focus on transaction-level anomaly detection in Section 2 in the revised paper.
>
> [1] Blockchain Large Language Models
>
> [2] Detecting Anomalies in Blockchain Transactions using Machine Learning Classifiers and Explainability Analysis
>
> [3] Anomaly Detection in Blockchain Networks: A Comprehensive Survey
>
> > **Reviewer's Comment:** The paper needs to clearly define what is an anomalous transaction. The ground truth for such anomalies is not clear. Only with a known ground truth of anomalous transactions can we discuss whether "using the reconstruction errors as the metric for anomaly detection" is appropriate.
>
> **Response:** Thank you for your question regarding the definition of anomalous transactions and the ground truth for these anomalies. In our study, we define anomalous transactions as those exhibiting behaviors that deviate significantly from typical transaction patterns, often due to logic or implementation flaws in smart contracts. Examples include transactions with abnormal method calls or sequences that differ significantly from expected patterns in benign transactions.
>
> For the ground truth, we have collected a set of known real-world malicious transactions, verified through reputable sources, including sources like CertiK and PeckShield. These transactions serve as the basis for defining anomalies, providing a clear ground truth to evaluate the model’s effectiveness.
>
> Using reconstruction errors as the metric for anomaly detection is appropriate here, as it captures deviations from learned benign patterns. Anomalous transactions tend to have higher reconstruction errors due to their irregularity, which aligns with our detection approach. We have further clarified this point in Section 4.1 in the revised paper.
>
> > **Reviewer's Comment:** There is a need for more detailed analysis on the reliability of the datasets used (types, the number of training samples, the number of testing samples, etc.). For example, the statement "our Ethereum dataset consists of 3,383 benign transactions for training, 709 benign transactions for testing, and 10 malicious transactions. The data was collected from October 2020 to April 2023." raises questions. Does this imply there were only ten malicious transactions on Ethereum from October 2020 to April 2023, or were these ten selected by the authors? If selected, what were the criteria for their selection?
>
> **Response:** Thank you for your question regarding our dataset. We apologize for any confusion. The data we collected focuses on specific DeFi applications within the Ethereum network, rather than all transactions on the entire Ethereum blockchain. The ten malicious transactions included were drawn from 5 DeFi applications and represent recorded and verified attack instances within that context.
>
> Our choice of these transactions is based on their availability from trusted sources that confirm their malicious nature. Due to the limited number of recorded and verified attacks specific to these applications, our dataset reflects the available data within this scope. We have highlighted this selection criterion in Section 5.1 and Section 6 in the revised paper.

---

> > ### Author Response · Authors · 2024-11-19
> >
> > > **Reviewer's Comment:** Why did the study use two different model architectures for different datasets? Does this indicate that the learning models for malicious transaction patterns lack transferability, necessitating the independent training of a model for each specific blockchain/dataset?
> >
> > **Response:** Thank you for this question. We selected different model architectures—BERT-base for the Ethereum task and BERT-large for the Solana task—primarily due to differences in the size of collected training data. Since we gather transaction data from DeFi applications, the volume of collectable transactions varies by blockchain. For instance, we sampled 3,383 benign transactions for Ethereum and 35,115 transactions for Solana. Using a BERT-large model on the Ethereum data would risk overfitting due to the smaller dataset, potentially degrading performance. We have added experimental results in our revised paper to support this choice and demonstrate the impact of model size on performance.
> >
> > > **Reviewer's Comment:** While the paper is clear and logically presented, it is more like a technical report than an academic paper.
> >
> > **Response:** Thank you for acknowledging the clarity and logical presentation of our paper. While our work is application-focused, we have made significant research contributions by customizing the masked language model for blockchain data. Specifically, we integrated FlashAttention and RoPE to improve performance and capture long-context dependencies, designed a multi-modal tokenization approach tailored to the unique characteristics of transaction data and developed a post-training detection solution. Our experiments demonstrate the effectiveness of these innovations in handling complex blockchain data. We design a new model and provide insights for our design choices. In addition, we analyze our experiment results and also design ablation studies to verify our design. These efforts are beyond the typical effort and depth of a technical report, which typically just runs existing techniques and shows the results.
> >
> > We believe this work goes beyond a technical report, as it provides a valuable study on applying transformer models to domain-specific data. We hope our approach will inspire further research on adapting language models for specialized fields such as blockchain and other security domains.

---

> > ### Comment · Reviewer_NnUh · 2024-11-22
> >
> > 1. You mentioned “ By detecting such anomalies, our approach enables transaction emergency stop, which can help prevent losses when malicious behavior is detected.” How to stop a transaction on a blockchain?
> >
> > 2. If there are only 10 malicious transactions in the dataset, will the class imbalance lead to issues with model performance evaluation, such as overfitting?

---

> > > ### Author Response · Authors · 2024-11-22
> > >
> > > Thank you for your insightful questions and feedback.
> > >
> > > 1.  **Emergency Stop of Transactions:** We apologize for the miscommunication in our previous response. To clarify, the "emergency stop" refers to halting specific contracts, such as suspending potentially risky withdrawal or lending services, rather than stopping individual transactions on the blockchain. We have revised the paper to reflect this clarification. Thank you for bringing this to our attention.
> > >
> > > 2.  **Class Imbalance and Overfitting:** Regarding the concern about the 10 malicious transactions, we want to emphasize that our model is trained exclusively on benign transactions to learn their common patterns. Malicious transactions are only used during testing to evaluate the model’s ability to detect anomalies based on deviations from the learned benign patterns. Thus it does not have the risk of overfitting to malicious transactions, as they are not part of the training process. We hope this clarification addresses your concerns.
> > >
> > > If you have any remaining questions or require further clarification, we would be more than happy to provide additional details.
> > >
> > > Thank you once again for your time and constructive feedback!

---

### Official Review · Reviewer_im5B · 2024-11-02

**Soundness:** 3
**Presentation:** 3
**Contribution:** 3
**Rating:** 6
**Confidence:** 3

**Summary:**

This paper presents BlockFound, a foundational model customized for detecting anomalies in blockchain transactions. It utilizes a modular tokenizer to handle multimodal inputs and employs masked language learning, RoPE embeddings, and FlashAttention technology for processing long sequence data. Extensive evaluations of transaction data from Ethereum and Solana demonstrate BlockFound's exceptional capabilities in anomaly detection while maintaining a low false positive rate.

**Strengths:**

1. BlockFound is a foundational model tailored for detecting anomalies in blockchain transactions, designed to accommodate their unique data structures.
2. A modular tokenizer processes multimodal inputs—specific tokens, text, and numbers—enhancing the accuracy of transaction feature capture.
3. Utilizing masked language learning, RoPE embeddings, and FlashAttention, the model effectively handles long sequence data, improving its ability to process lengthy transaction records.
4. Experimental evaluations on Ethereum and Solana transactions show that BlockFound excels in anomaly detection, demonstrating high accuracy and low false positive rates.
5. Design for the DeFi environment, offering valuable detection methods to protect user assets and enhance the security of blockchain financial transactions.

**Weaknesses:**

1. The paper primarily conducts experimental evaluations on Ethereum and Solana networks, lacking validation of the model's generalization capability and adaptability.
2. The paper highlights the model's performance but lacks transparency in decision-making. Integrating tools could improve interpretability and enhance user trust.
3. The BlockFound model may need further adjustments to meet privacy standards.

**Questions:**

1. How does fine-tuning the GPT-4 model using the API enhance its performance for the same tasks or domains?
2. Can the model's generalization capability be further validated on other blockchain platforms?
3. As the blockchain ecosystem evolves, new attack patterns and transaction types may emerge. Can the model adapt to these changes?
4. Does noise and inaccurate information in the data significantly impact the model's performance?
5. In handling blockchain transaction data, privacy protection is a crucial consideration. How does the article address and safeguard user privacy data?

---

> ### Author Response · Authors · 2024-11-19
>
> > **Reviewer's Comment:** The paper primarily conducts experimental evaluations on Ethereum and Solana networks, lacking validation of the model's generalization capability and adaptability.
>
> **Response:** Thank you for emphasizing the importance of generalization and adaptability across blockchain platforms. We agree that validating our model on diverse platforms is valuable. In this work, we selected Ethereum and Solana as representative platforms due to their widespread usage and differing architectural characteristics, which allow us to demonstrate our model's applicability across varied blockchain types. Given the non-trivial nature of data collection for additional platforms, we have focused on these two as a starting point and will consider expanding to other platforms in future work. We have also added this point to our discussion section in the revised paper.
>
> > **Reviewer's Comment:** The paper highlights the model's performance but lacks transparency in decision-making. Integrating tools could improve interpretability and enhance user trust.
>
> **Response:** Thank you for this valuable suggestion. We agree that interpretability is important, as it can help enhance user trust and provide greater transparency in the model’s decision-making process. In this paper, our primary focus is on establishing effective models rather than building new interpreting methods for foundation models; however, we recognize the importance of incorporating interpretability tools and have highlighted this point to our discussion section as an area for future work.
>
> > **Reviewer's Comment:** In handling blockchain transaction data, privacy protection is a crucial consideration. How does the article address and safeguard user privacy data?
>
> **Response:** Thank you for this important question. We are committed to upholding data privacy standards. In our work, we utilize only publicly available transaction data from blockchain networks, which are fully transparent and downloadable by design. No private or permissioned blockchain data is used. Additionally, user addresses in these transactions are represented as hashed identifiers, inherently protecting user-sensitive information and making it impossible to trace back to personal identities. This ensures that our research respects and safeguards user privacy. We have highlighted this point in our discussion section in the revised paper.
>
> > **Reviewer's Comment:** How does fine-tuning the GPT-4 model using the API enhance its performance for the same tasks or domains?
>
> **Response:** Thank you for this suggestion. We conducted experiments fine-tuning GPT-4 (version 2024-08-06) using the Ethereum dataset to learn benign transaction patterns. Our approach was inspired by BlockGPT but with modifications due to the constraints of OpenAI’s fine-tuning API. Specifically, we used the first half of each benign transaction as input and the last half as the target response, enabling the model to learn transaction prediction. The error between the predicted and actual transaction is used as the anomaly score.
>
>
>
> Unlike BlockGPT’s token-by-token iteration approach, we opted for this method because:
>
> 1.  The current OpenAI fine-tuning API supports only instruction-response style fine-tuning, not token-by-token iteration.
>
> 2.  Generating $N−1$ training samples for a transaction with $N$ tokens would result in prohibitively high costs. Even with our modified approach, the fine-tuning and inference process cost approximately $950.
>
>
> Our results, included in Appendix C.2 in the revised paper, show no significant improvement compared to using GPT-4o directly without fine-tuning. Possible reasons include:
>
> 1.  The model might require more training iterations than our budget allowed.
>
> 2.  The half-half prediction approach might be too coarse-grained to capture fine transaction details.
>
> 3.  The default tokenization might make it difficult for GPT-4 to handle specific patterns, such as addresses and numbers.
>
>
> Further exploration of these issues would require greater control over GPT-4 fine-tuning and tokenization, which is currently not feasible. However, we believe a more fine-grained fine-tuning approach, potentially integrating tool usage, could improve performance. We have added this as a future direction in our discussion section.

---

> > ### Author Response · Authors · 2024-11-19
> >
> > > **Reviewer's Comment:** As the blockchain ecosystem evolves, new attack patterns and transaction types may emerge. Can the model adapt to these changes?
> >
> > **Response:** Thank you for this insightful question. Since our model is trained solely on benign data, it learns typical transaction patterns and detects anomalies based on deviations from these patterns. This approach provides a certain level of adaptability to new or previously unseen malicious strategies, as evidenced by our results in Section 5.1, where our model successfully detects anomaly transactions that occurred after the benign transaction sampling period. However, if new attack strategies closely mimic benign patterns, detection may indeed be challenging.
> >
> > Furthermore, as the blockchain ecosystem evolves, the transaction distribution may diverge significantly from the data used for training. To address this, we recommend periodically retraining the model on updated data to ensure optimal detection accuracy. We have included this discussion in the revised paper and have reported the training overhead, which remains manageable due to FlashAttention’s integration, keeping the time cost minimal. We have added the discussion in Section 6 in our revised paper.
> >
> > > **Reviewer's Comment:** Does noise and inaccurate information in the data significantly impact the model's performance?
> >
> > **Response:** Thank you for raising this important question. In blockchain transactions, the ratio of benign to malicious samples is typically highly imbalanced (e.g., 1,000:1). Given this imbalance, even if some potential malicious samples are inadvertently included in the training set, their impact is minimal, as the model predominantly learns the representation of the majority class (benign transactions).
> >
> > To assess the model's robustness to noise, we conducted an experiment simulating an extreme case where half of the malicious transactions were intentionally included in the training set for the Ethereum dataset. While this scenario caused a slight drop in detection performance, the model remained effective, achieving a recall of 60% for a detection threshold $k=10$. These results demonstrate that our approach maintains robustness to a reasonable level of noise and inaccurate information in the data.
> >
> > We have added these findings and related discussions in Section 6 and Appendix C.2 to the revised paper to highlight the model’s resilience.

---

> > > ### Author Response · Authors · 2024-11-22
> > >
> > > Thank you again for the review. Your careful reading and insightful comments indeed help us a lot in improving our work.
> > > Since the discussion phase is about to end, we are writing to kindly ask if the reviewer has any additional comments regarding our response. In addition, if our new experiments address your concern, we would like to kindly ask if you could consider raising the score.

---

> > > > ### Comment · Reviewer_im5B · 2024-11-26
> > > >
> > > > I would like to thank the authors for their response. However, upon further reflection, I have concerns about the validity and persuasiveness of the problem that the paper addresses. I will keep my current rating.

---

> > > > > ### Author Response · Authors · 2024-11-26
> > > > >
> > > > > Thank you for keeping a positive score and for your careful review and instructive feedback. We truly appreciate the time and effort you have dedicated to evaluating our work.
> > > > >
> > > > > We would like to take this opportunity to clarify the validity and persuasiveness of the problem our paper addresses: Blockchain anomaly detection is a critical area given the increasing adoption of decentralized financial systems and the growing risks associated with malicious transactions. Our work specifically targets anomalies arising from smart contract vulnerabilities, which represent the majority of real-world attacks in this domain. By leveraging transformer-based architectures and tailored adaptations, we provide a novel and practical approach to addressing these challenges.
> > > > >
> > > > > Since the ICLR discussion period has been extended, if you have any new concerns or additional questions, please do not hesitate to let us know. We are more than willing to provide further clarifications or address any remaining points.
> > > > >
> > > > > Thank you again for your valuable input and constructive engagement.

---

### Official Review · Reviewer_h1P4 · 2024-11-03

**Soundness:** 2
**Presentation:** 2
**Contribution:** 2
**Rating:** 5
**Confidence:** 4

**Summary:**

This paper proposes a deep-learning model for anomaly blockchain transaction detection. By designing a modularized tokenizer for blockchain-specific tokens, texts, and numbers in the transactions, the model handles these multi-modal inputs, balancing the information across different modalities. The model also adapts the RoBERT to train the foundation model with a mask language learning, and makes use of the encoder to detect anomaly transactions.
The evaluation of the model on a meticulously curated dataset demonstrates its potential for applications, showcasing practical applicability in specific environments.

**Strengths:**

1.The paper designs a novel tokenizer that deal with different types of multi-modal inputs in a transaction record, which capture the sematic in different fields effectively and shows some performance improvement;
2.The paper adopts BERT-like model and use
mask language modeling to train the foundation model rather than using GPT-style models, forming a lighter model architecture.
3.The paper evaluates the proposed model in real-world transaction data, and the results appear to be quite promising, especially on Solana with high accuracy.

**Weaknesses:**

1.Although the adapted foundation model shows some performance improvement with RoPE and FlashAttention, it represents a typical data mining method that has been extensively applied to various tasks, thus limiting its originality.

2.The authors rank all the addresses in transactions by frequency and retain the top 7,000 most frequent addresses. Obviously, some information is lost, and there is no guarantee that low-frequency addresses and transactions are necessarily benign.

3.FlashAttention is a key module to handle long inputs in the foundation mode. However, there has been no experimental evidence that this approach has any impact on the accuracy or complexity of the model.

4.There are several Embeddings as inRoBERT such as Position Embedding + Token Embedding and Segment Embedding. However, the authors does not describe how many embeddings are adopted and what is the difference between these embeddings used in their model and those we generally known in RoBERT.

5.The authors should clarify why specific methods, such as RoBERT, were chosen for the framework since there are a number of other BERT-like models that can be used for the modelling of long sequences, such as ELECTRA.

6.Some seem too subjective as no relevant cases or references were found, for example: “..these models cannot capture
the long-range dependencies and complex temporal dynamics inherent in transaction data, resulting sub-optimal modeling performance...” in Section I and "...using MLM can significantly reduce
the computational cost...." in Section III.B

**Questions:**

1.What is the time and space complexity of the proposed method, and how does FlashAttention affect it?

2.Are low-frequency addresses BENIGN? How well does the model work if low frequency addresses are retained?

3.Why does the model chose RoBERT as foundation model and not another model?

4.In Tokenizer, all tokens share the same vocabulary, since unique hash addresses are individual tokens, how can the size of this vocabulary be controlled to make it available at all times?

---

> ### Author Response · Authors · 2024-11-19
>
> We thank the reviewer for the careful reading and valuable suggestions. Below are our responses:
>
> > **Reviewer's Comment:** Although the adapted foundation model shows some performance improvement with RoPE and FlashAttention, it represents a typical data mining method that has been extensively applied to various tasks, thus limiting its originality.
>
> **Response:** Thank you for the comment. We agree with the reviewer that RoPE and FlashAttention have been widely used in training foundation models. In this work, we adopt them to train the SOTA foundation model for blockchain transactions. We believe they are reasonable design choices because they help address our key challenge in blockchain transaction foundation model development: long sequence.
>
> We would like to kindly point out that our main technical contribution is the introduction of multi-modal tokenization and a post-training detection approach. The multi-modal nature of transaction data introduces significant complexity, and our tokenization technique is specifically tailored to manage this, ensuring the model effectively interprets distinct transaction features. Furthermore, the post-training detection step allows our model to identify anomalies within a transaction context effectively.
>
> To the best of our knowledge, our work is the first to combine advanced model construction techniques with our customized designs for training blockchain foundation models. Given our customized designs and the effectiveness of the trained model, we respectfully point out that applying RoPE and FlashAttention does not dilute our contribution.
>
> > **Reviewer's Comment:** In Tokenizer, all tokens share the same vocabulary, since unique hash addresses are individual tokens, how can the size of this vocabulary be controlled to make it available at all times?
>
> **Response:** Thank you for your question. In our approach, we initially build the tokenizer using a large transaction corpus. For unique addresses, we rank them by frequency and retain the most common ones (e.g., the top 7,000) as unique tokens, while replacing less frequent addresses with an [OOV] token. We acknowledge that this approach has limitations; if the current data distribution diverges significantly from the distribution used to create the tokenizer, a high number of [OOV] tokens could potentially degrade performance. To mitigate this risk, we recommend that developers periodically rebuild the tokenizer, either after a specific number of transactions or when a noticeable shift in data distribution occurs. This will help maintain optimal detection performance by keeping the tokenizer aligned with current transaction patterns. Thanks for pointing this out and we have added this to our discussion in Section 6 in the revised paper.
>
> > **Reviewer's Comment:** FlashAttention is a key module to handle long inputs in the foundation mode. However, there has been no experimental evidence that this approach has any impact on the accuracy or complexity of the model.
>
> **Response:** Thank you for raising this point. We have conducted an ablation study, which we have included in the revised paper, to evaluate the impact of FlashAttention on both training speed and memory efficiency. For Ethereum, FlashAttention reduces the running time from 9,415 seconds to 7,024 seconds and nearly halves GPU memory usage. As expected, the accuracy remains similar, as FlashAttention primarily focuses on optimizing attention computation by leveraging IO-awareness through GPU SRAM, as outlined in the original FlashAttention paper.
>
> For Solana, the significance of FlashAttention is even more evident. Without FlashAttention, even a batch size of 1 cannot be processed on an 80GB A100 GPU due to memory constraints, while FlashAttention enables efficient training with a batch size of 2. These results demonstrate the critical role of FlashAttention in handling long sequences and enabling feasible training for large-scale datasets. We have updated Appendix C.3 in the paper to reflect these findings.

---

> > ### Author Response · Authors · 2024-11-19
> >
> > > **Reviewer's Comment:** The authors rank all the addresses in transactions by frequency and retain the top 7,000 most frequent addresses. Obviously, some information is lost, and there is no guarantee that low-frequency addresses and transactions are necessarily benign.
> >
> > **Response:** Thank you for your thoughtful questions. For Ethereum, there are 7,335 unique addresses in total, and for Solana, there are 56,203 unique addresses. We retain the top 7,000 unique addresses based on frequency to balance training efficiency and information retention. For Ethereum, this covers the majority of addresses, leaving only those that appear once in the training set. For Solana, retaining all addresses would significantly increase the embedding size, making training computationally infeasible given the extensive training resources required (We set batch size as only 2 for Solana even with FlashAttention). Importantly, the addresses beyond the top 7,000 appear fewer than 10 times in the training set, contributing minimally to overall information.
> >
> > Regarding whether low-frequency addresses are benign, high-frequency addresses in blockchain datasets often correspond to commonly used smart contracts, token addresses, or other frequently accessed entities. Low-frequency addresses, such as user wallet addresses, typically carry less significance for classification tasks. Their exclusion does not adversely impact the model’s performance since they contribute minimal useful patterns for detecting anomalies. Moreover, retaining low-frequency addresses would increase model complexity and training time without adding significant value.
> >
> > By focusing on the top 7,000 addresses, we optimize the trade-off between retaining sufficient information for effective anomaly detection and ensuring training efficiency. We have added this discussion in Appendix B to the revised paper.
> >
> > > **Reviewer's Comment:** There are several Embeddings as inRoBERT such as Position Embedding + Token Embedding and Segment Embedding. However, the authors does not describe how many embeddings are adopted and what is the difference between these embeddings used in their model and those we generally known in RoBERT.
> >
> >
> >
> > **Response:** We appreciate the reviewer’s question regarding the embeddings in our model. To clarify, we replace the absolute positional embeddings used in RoBERTa with RoPE to better handle long-sequence transaction data. For token embeddings and segment embeddings, we retain the standard design from RoBERTa, using these embeddings in the same way to train our model from scratch. This approach allows us to leverage RoPE’s advantages for positional encoding while maintaining the proven effectiveness of RoBERTa’s token and segment embedding structures.
> >
> > > **Reviewer's Comment:** The authors should clarify why specific methods, such as RoBERT, were chosen for the framework since there are a number of other BERT-like models that can be used for the modelling of long sequences, such as ELECTRA.
> >
> > **Response:** Thank you for the suggestion. We chose RoBERTa as it is one of the state-of-the-art methods in the BERT series and has been widely adopted for various downstream tasks. However, we would like to emphasize that our framework is agnostic to the choice of the base model. To validate this, we conducted additional experiments using other state-of-the-art models, such as DeBERTa, on the Ethereum dataset. The results were consistent with those achieved using RoBERTa, demonstrating that the choice of base model does not significantly affect performance in our framework.
> >
> > We have updated the revised paper to include these experimental results and discuss the potential of extending our evaluation to other models as part of future work in Appendix C.3 in our revised paper.

---

> ### Author Response · Authors · 2024-11-19
>
> > **Reviewer's Comment:** Some seem too subjective as no relevant cases or references were found, for example: “..these models cannot capture the long-range dependencies and complex temporal dynamics inherent in transaction data, resulting sub-optimal modeling performance...” in Section I and "...using MLM can significantly reduce the computational cost...." in Section III.B
>
> **Response:** Thank you for pointing this out. We have revised the paper to include relevant references and correct the claim we made. To clarify:
>
> 1.  The advantage of transformers over LSTMs in capturing long-range dependencies has been well-documented in prior studies [1, 2, 3], which we have now cited in the revised paper.
>
> 2.  Regarding the computational cost of using MLM versus GPT-style models, we appreciate you bringing this to our attention. After re-evaluating our statements, we recognize that our previous claims regarding the computational cost differences between MLMs and GPT-style models were inaccurate. For the anomaly detection task, with the given sequence, we can input the full sequence into the model in a single forward pass to compute the hidden states and output logits for each position in parallel. Thus, the time complexity for both MLM and GPT-style models is $O(N^2)$. We have updated the manuscript to correct this oversight. Our primary objective is to learn meaningful representations of transaction patterns to identify anomalies effectively, rather than to generate new sequences. While GPT-style models are powerful for generative tasks, MLMs offer an effective framework for encoding input data without the necessity of learning sequence generation. This makes MLMs particularly well-suited for our anomaly detection task.
>
>
> We have included the reference and the correct the claim in Section 1 of the revised paper to clarify these points. Thank you again for your valuable feedback, which has helped us improve the accuracy and clarity of our paper.
>
> [1] Stabilizing transformers for reinforcement learning
>
>
>
> [2] A comparison of transformer and lstm encoder decoder models for asr
>
>
>
> [3] Transformers in Time Series: A Survey

---

> > ### Author Response · Authors · 2024-11-22
> >
> > Thank you again for the review. Your careful reading and insightful comments indeed help us a lot in improving our work.
> > Since the discussion phase is about to end, we are writing to kindly ask if the reviewer has any additional comments regarding our response. In addition, if our new experiments address your concern, we would like to kindly ask if you could consider raising the score.

---

### Official Review · Reviewer_3gbo · 2024-11-04

**Soundness:** 2
**Presentation:** 3
**Contribution:** 3
**Rating:** 6
**Confidence:** 4

**Summary:**

The paper proposes BlockFound, a customized foundation model for anomaly transaction detections in Blockchain. Traditional rule-based approaches and techniques relying on off-the-shelf language models struggle with generalizability and scalability. In order to address previous limitations, BlockFound leverages a multi-modal tokenizer to handle the transaction features along with a Bert-based architecture using mask language modeling (MLM) to reduce computational demands. The experiment results demonstrate BlockFound outperforms the baseline models by achieving higher accuracy with lower false positive rates.

**Strengths:**

- Clear justification for using multi-modal approach to process input Blockchain transactions.

- Comprehensive evaluation between BlockFound and baseline models with ablation studies for validating the effectiveness of the designs.

- The implementation of the model and data are open-sourced.

**Weaknesses:**

- The definition of anomaly transactions is ambagious.

- The size of malicious transaction data is limited.

- Potential redundancy in transaction data. Since contract templates (e.g., ERC-20 tokens) are wildly used in smart contract development, the dataset may contain duplicate transactions.

**Questions:**

I appreciate authors’ efforts in identifying anomaly transactions in Blockchain via machine learning-based techniques. The methodology is clear and the model design is persuasive in general, but I still have several concerns on the transaction data used in this manuscript along with questions for authors to respond.

## Q1: The definition of anomaly transactions is ambagious.
Since the blockchain transactions can vary wildly in structure and content, it would be helpful to have a clear definition of the anomaly transactions. From my understanding, there could be various kinds of transactions regarded as anomaly. For example, the transactions emitted by malicious users to trigger smart contract vulnerabilities either on the implementation level (e.g., integer overflow and reentrancy) or logic level (e.g., inappropriate access control to methods) can be one kind of anomaly transactions. The fraud/phishing transactions belong to another kind. Could the authors supplement a clear definition of the anomaly transactions studied in this paper?

## Q2: The size of malicious transaction data is limited.

I noticed that only 28 malicious transactions in total were used in the training/testing dataset, which is quite limited. I am concerned about if the model can effectively learn the features of anomaly transactions from such a small dataset. Could the authors justify how the proposed model can reliably learn and differentiate the benign and malicious transactions based on the imbalanced dataset?

## Q3: Potential redundancy in transaction data.

Contract templates are wildly used in real-world smart contract development, leading to different smart contracts may offering similar or even identical APIs to the users. This could cause potential duplication in the transaction data. I have two related questions for the authors:

- Q3.1: For the transactions calling to the same methods with different argument values, do they contribute the equally to model training, i.e., can they be considered as duplicate?

- Q3.2: If the above transactions are indeed duplicate, is it possible that similar redundant transactions are present in the dataset used for this study?

---

> ### Author Response · Authors · 2024-11-19
>
> We thank the reviewer for the careful reading and valuable suggestions. Below are our responses:
> > **Reviewer's Comment:** The definition of anomaly transactions is ambagious.
>
> **Response:** We thank the reviewer for highlighting the need for a clearer definition of anomaly transactions. Here, anomaly transactions refer to transactions that violate the design logic and usage of smart contracts and DeFi protocols. These anomaly transactions happen when attackers exploit the vulnerabilities in smart contracts and will cause serious financial losses. Given that different smart contracts may have different types of vulnerabilities, the anomaly transaction patterns will also be different. In our paper, we share a similar goal to the prior work [1] and focus on identifying transactions exhibiting abnormal behaviors in comparison to benign transactions within the selected DeFi applications. We have clarified this definition in Section 2 in the revised paper.
>
> [1] Blockchain Large Language Models
>
> > **Reviewer's Comment:** The size of malicious transaction data is limited.
>
> **Response:** We appreciate the reviewer’s concern about the dataset size. We would like to clarify that our model is trained exclusively on benign data to learn typical transaction patterns. During testing, the model identifies malicious transactions by detecting deviations from these learned benign patterns.
>
> Regarding the sufficiency of 28 malicious transactions for evaluation, we acknowledge the challenges posed by this limited dataset. However, as detailed in Section 5.1, these transactions originate from well-established DeFi applications and have been carefully sampled from reputable sources, like CertiK and PeckShield. Each transaction was manually verified to confirm its malicious nature. They are real-world malicious transactions exploiting the smart contract vulnerability for the selected DeFi apps, and thus the number can be limited. Moreover, in our work, we utilize only publicly available transaction data from blockchain networks, and no private or permissioned transactions are used to make the dataset meet the privacy standard and able to open-source. Collecting a larger set is non-trivial; given the amount of manual effort needed for confirming them. Our future work will continue expanding the dataset. Note that prior work [1] identified a total of 116 malicious transactions, yet they were unable to share these transactions or their sources with us due to privacy concerns. We have added this discussion in Section 6 in the revised paper and open-sourced our datasets and models to foster further research.
>
>
> > **Reviewer's Comment:** Contract templates are wildly used in real-world smart contract development, leading to different smart contracts may offering similar or even identical APIs to the users. This could cause potential duplication in the transaction data.
>
> **Response:** Thank you for raising this important point. To assess potential duplication in the dataset, we conducted a 5-gram BLEU similarity analysis (We chose 5-gram to avoid the false positives lead by indicator tokens e.g., [START], [CALL]). In the Ethereum dataset, only 0.05% of transaction pairs show a BLEU similarity over 0.7, and 0.023% exceed 0.8. These highly similar transactions may indeed result from contract templates. Given the low similarity ratio in our data, we do not consider potential duplication a significant issue. We have updated this discussion in Appendix B in our revised paper.

---

> > ### Author Response · Authors · 2024-11-22
> >
> > Thank you again for the review. Your careful reading and insightful comments indeed help us a lot in improving our work.
> > Since the discussion phase is about to end, we are writing to kindly ask if the reviewer has any additional comments regarding our response. In addition, if our new experiments address your concern, we would like to kindly ask if you could consider raising the score.

---

> > > ### Comment · Reviewer_3gbo · 2024-11-27
> > >
> > > Thanks for answering my questions! I appreciate authors' efforts in clarifying the definition of abnormal transactions and the possible duplicate smart contracts in the dataset.
> > > However, I still have concerns over the limited size of abnormal transactions in the used dataset.
> > > Thus I will keep my current rating.

---

> > > > ### Author Response · Authors · 2024-11-27
> > > >
> > > > Thank you for your response and for acknowledging that we have addressed some of your concerns. We appreciate the opportunity to clarify the remaining concern regarding the limited size of abnormal transactions in our dataset.
> > > >
> > > > Firstly, we would like to emphasize that these abnormal transactions are not used during training; they are **solely utilized for testing purposes**. It makes the anomaly detection challenging considering the size. For example, in the Solana dataset, the detector needs to recall 18 abnormal transactions from a pool of 1,500 transactions. The high recall rate achieved demonstrates the effectiveness of our method, as the detector has successfully learned the representation of benign transaction patterns and identified deviations indicative of anomalies.
> > > >
> > > > Moreover, collecting these abnormal transactions is a non-trivial task. We focus on public transactions from selected DeFi applications, and each transaction is manually verified to confirm its malicious nature. This effort ensures the integrity and reliability of our dataset, and we have open-sourced it to foster further research in this area. As a point of comparison, BlockGPT collected 116 malicious transactions but has not released them or the source from which they were collected.
> > > >
> > > > Expanding the dataset is indeed an important next step, but it requires substantial additional effort, particularly in verifying transaction maliciousness. We have acknowledged this in the paper and identified it as a direction for future work.
> > > >
> > > > Please let us know if this clarification could address your concern.

---

> > > > > ### Comment · Reviewer_3gbo · 2024-11-27
> > > > >
> > > > > Thanks for your clarification. I have updated my score correspondly.

---

> > > > > > ### Author Response · Authors · 2024-11-27
> > > > > >
> > > > > > Thanks for raising the score! We appreciate your recognition and support, your feedback helps us improve our work!

---

### Author Response · Authors · 2024-11-19
**Rebuttal Summary**

**Rebuttal Summary**

We thank the reviewers for their constructive feedback. Below, we summarize the new experiments and changes made in the revised paper:

----------

### **New Experiments**
1.  **Impact of FlashAttention (Reviewer h1P4)**

    -   Conducted an ablation study to evaluate training time and memory efficiency with and without FlashAttention.
    -   Results: FlashAttention reduced training time (e.g., Ethereum: 9,415s → 7,024s) and nearly halved GPU memory usage, enabling training for Solana.
2.  **Choice of RoBERTa (Reviewer h1P4)**

    -   Conducted experiments with DeBERTa to validate that the choice of base model has minimal impact on performance, emphasizing framework agnosticism.
3.  **Fine-tuning GPT-4o (Reviewer im5B)**

    -   Performed fine-tuning on GPT-4o using Ethereum data to assess anomaly detection performance. Results showed limited improvement over direct GPT-4 use, highlighting constraints of the current fine-tuning API.
4.  **Evaluation of noisy data (Reviewer im5B)**

    -   Simulated the inclusion of malicious samples in the benign training set and conducted an ablation study to assess robustness under noise rates of 50%
    -   Added results to show that the model maintains reasonable performance despite noise, demonstrating resilience.
----------
Below, we also summarize the key points in our responses:

### Key Points in Our Responses

**Reviewer 3gbo**

1.  We clarified the definition of anomalous transactions, emphasizing their relevance to deviations from typical patterns and their potential to cause financial losses.
2.  We clarified the concerns about the dataset size, explaining the careful selection of verified malicious transactions from reputable sources like CertiK and PeckShield.
3.  We analyzed potential redundancy in the transaction data and demonstrated that duplication is minimal, with only 0.05% of transactions showing high BLEU similarity.

----------

**Reviewer h1P4**

1.  We justified the use of RoPE and FlashAttention, explaining their necessity for handling long transaction sequences, and highlighted our main technical contribution is the introduction of multi-modal tokenization and a post-training detection approach.
2.  We clarified tokenization strategies to highlight the trade-offs between computational efficiency and information retention.
3.  We conducted additional experiments to evaluate the impact of FlashAttention on time and memory efficiency, showing significant improvements.
4.  We clarified the embedding structures used in our approach.
5.  We demonstrated that the choice of base model, like DeBERTa or RoBERTa, does not significantly affect framework performance, supporting model-agnostic applicability.

----------

**Reviewer im5B**

1.  We discussed the generalization concerns, explaining the selection of Ethereum and Solana networks and our future work.
2.  We emphasized our commitment to data privacy, using only public blockchain data and hashed identifiers to safeguard sensitive information.
3.  We discussed interpretability as a key future direction to enhance user trust and transparency, leaving it as future work.
4.  We demonstrated the model’s adaptability to evolving blockchain ecosystems by suggesting periodic retraining and validating its performance on post-training data.

----------

**Reviewer NnUh**

1.  We added references in the introduction to elaborate on the motivation for anomaly detection in blockchain transactions.
2.  We clarified the definition of anomalous transactions and their ground truth, explaining the verification process for malicious transactions used in the dataset.
3. We clarified the malicious transaction selection process.
4.  We clarified the differences in model architectures for Ethereum and Solana datasets, explaining the rationale behind these choices based on dataset size and overfitting risks.
5.  We highlighted our contribution as a research paper in studying anomaly detection for blockchain transactions.

---

### Author Response · Authors · 2024-12-03
**Rebuttal and Discussion Summary**

Dear Reviewers and AC,

We sincerely thank all the reviewers for their insightful comments and recognition of our work. We are encouraged that the reviewers have highlighted the following strengths in our paper:

-   **Innovative approach to blockchain anomaly detection using large language models** (Reviewers im5B, NnUh)
-   **Effective use of a multi-modal tokenizer** to process blockchain transactions, capturing semantics across different modalities (Reviewers 3gbo, h1P4, im5B)
-   **Comprehensive evaluation** with real-world data and ablation studies validating the effectiveness of our designs (Reviewers 3gbo, h1P4, NnUh)
-   **Successful adaptation of transformer architectures**, including RoPE embeddings and FlashAttention, to handle long transaction sequences (Reviewers h1P4, im5B)
-   **Promising results in anomaly detection** with high accuracy and low false positive rates, especially on the Solana dataset (Reviewers h1P4, im5B, NnUh)

Below, we list the main concerns raised by the reviewers and highlight how we have addressed them during the discussion period.

----------

**Reviewer 3gbo:**

-  We clarified the definition of anomalous transactions in our setting and emphasized that our model is trained solely on benign data, with malicious transactions used only for testing.
----------

**Reviewer h1P4:**

-   We highlighted that our main technical contributions include multi-modal tokenization and post-training detection. While the reviewer did not explicitly respond, we believe our explanation adequately addressed their concerns.

-   We conducted an ablation study showing significant improvements in training time and memory efficiency with FlashAttention, which was included in the revised paper.

-    We explained the trade-offs between computational efficiency and information retention in retaining only the top 7,000 addresses.

-   We provided experimental evidence demonstrating that the framework's performance is agnostic to the choice of base model.

-   We revised the paper to include relevant references and corrected subjective claims, as suggested.

----------

**Reviewer im5B:**

-   We detailed experiments with fine-tuning GPT-4, highlighting limitations due to API constraints and costs. The reviewer acknowledged this response but maintained their original rating.

-  We conducted an ablation study simulating noisy data inclusion, demonstrating the model's resilience.

-   We clarified that only publicly available blockchain data was used, ensuring privacy protection.


----------

**Reviewer NnUh:**

-   We expanded the introduction and added references to support the significance of transaction-level anomaly detection.

-   We provided more details about the dataset, including criteria for malicious transaction selection and the rationale for dataset size. The reviewer raised a concern about class imbalance, which we addressed by explaining that malicious transactions were used only for testing.

-   We explained the use of different architectures for Ethereum and Solana due to differences in dataset size.

-   We highlighted the depth of our contributions and efforts to ensure the paper moves beyond a technical report.

----------

In conclusion, we have thoroughly addressed the reviewers’ concerns, with many points clarified and acknowledged during the discussion period. For concerns without explicit responses, we believe our detailed clarifications sufficiently addressed the issues raised. We hope this summary provides a clear overview of our efforts to engage constructively with the reviewers.

Thank you for your time and for overseeing this process.

Best regards,
Authors

---

### Meta-Review · Area_Chair_hgfy · 2024-12-20

**Metareview:**

This paper presents a deep learning approach for anomaly blockchain transaction detection: BlockFound. It proposed a modularized tokenizer to handle multimodal blockchain-specific elements in the transactions, balancing the information across different modalities. The approach trains a foundation model using a mask language modeling is used to reduce computational costs. A large experimental evaluation is provided.

Strengths:
- a new method deals with blockchain transactions,
- multi-modal approach,
- use of light BERT-like model and masked language learning,
- comprehensive and promising evaluation real-world data.

Weaknesses:
- originality limited,
- definition of anomaly transactions ambiguous,
- some aspects of the methods should be better discussed and evaluated (use of transactions with top-frequencies,  size of malicious transactions limited, presence of duplicate transactions)
-generalization and adaptability not well evaluated,
- privacy issues raised questions.


During the rebuttal, authors did a strong effort to provide specific answers to the remarks raised by reviewers, they also proposed additional experiments and a revision of the paper.
During the discussion, the reviewers highlighted the interest of the paper as a unique approach to blockchain transactions and that the authors have addressed some of reviewers' issues.  However, there was a clear consensus for saying that the paper lacks a depth analysis,  a theoretical support and should consider a restructuration.
While the paper has merits, they agreed that the paper is not ready in the proposed revision.

Following these elements, I propose rejection.
Nevertheless, I encourage the authors to improve their work for other venues.

**Additional Comments On Reviewer Discussion:**

During discussion, reviewer 3gbo -who increased his score to 5 after rebuttal- indicated that the paper is somehow below the bar, even though the authors justified parts of his concerns. He did not support acceptance.

Reviewer im5B mentioned that for his the paper lacks the depth and structure expected of an academic paper: It reads more like a technical report. He thinks that the paper is not currently acceptable for publication.

Reviewer NnUh indicated that for him the paper is clearly written as technical report and lacks theoretical support. He was not convinced by the definition and experiment of anomaly detection.

Reviewer h1P4, did not mention a support to the paper.

The scores of the 4 reviewers was 5, the consensus was the paper has some merits but the revision provided is below the bar.

---

### Decision · Program_Chairs · 2025-01-22

Reject